# Effectiveness of Dry Needling of Myofascial Trigger Points in the Triceps Surae Muscles: Systematic Review

**DOI:** 10.3390/healthcare10101862

**Published:** 2022-09-24

**Authors:** David Lucena-Anton, Carlos Luque-Moreno, Jesus Valencia-Medero, Cristina Garcia-Munoz, Jose A. Moral-Munoz

**Affiliations:** 1Department of Nursing and Physiotherapy, Faculty of Nursing and Physiotherapy, University of Cadiz, 11009 Cadiz, Spain; 2Department of Physiotherapy, Faculty of Nursing, Physiotherapy and Podiatry, University of Seville, 41009 Seville, Spain; 3Institute of Research and Innovation in Biomedical Sciences of the Province of Cadiz (INiBICA), University of Cadiz, 11009 Cadiz, Spain

**Keywords:** dry needling, trigger points, triceps surae, physical therapy

## Abstract

This study aims to analyze the effects of Dry Needling (DN) for the release of myofascial trigger points (MTrPs) in the triceps surae muscles (TSM). A systematic review was performed up to February 2022 in PubMed, PEDro, Scopus, CENTRAL, and Web of Science. Selection criteria were studies involving subjects older than 18 years presenting MTrPs in the TSM, without any concomitant acute or chronic musculoskeletal conditions; DN interventions applied to the MTrPs of the TSM; and results on pain, range of motion (ROM), muscle strength, muscle stiffness, and functional outcomes. The PEDro scale was used to assess the methodological quality of the studies, and the Risk of Bias Tool 2.0 to assess risk of bias. A total of 12 studies were included in the systematic review, involving 426 participants. These results suggest that DN of MTrPs in TSM could have a positive impact on muscle stiffness and functional outcomes. There are inconclusive findings on musculoskeletal pain, ROM, and muscle strength. Significant results were obtained in favor of the control groups on pressure pain thresholds. Despite the benefits obtained on muscle stiffness and functional performance, the evidence for the use of DN of MTrPs in the TSM remains inconclusive.

## 1. Introduction

Myofascial pain is a clinical syndrome derived from musculoskeletal pain, which presents with a referred component and is diagnosed by rigorous examination to locate myofascial trigger points (MTrPs) [1,2,3]. MTrPs are clinically defined as a hyperirritable nodule of spot tenderness located in a taut band of skeletal muscle which is tender and palpable through physical examination [4]. A recent study carried out by Li et al. [5] stated that the diagnosis of MTrPs is mostly based on the presence of three criteria, either stand-alone or combined: spot tenderness, referred pain and local twitch response (LTR). The use of these criteria combined provide a more reliable diagnosis, as it is known that the reliability of each criterion is associated with the analyzed muscle [6]. In addition to physical examination, taut bands can be objectively characterized by magnetic resonance [7], and the irritability caused by MTrPs can be showed by electromyography [3,8]. Nevertheless, an objective standard diagnostic of MTrPs is still needed [5].

The most common muscle dysfunctions that can be caused by these MTrPs are a decreased range of motion, weakness, fatigue, referred spasm, poor post-exercise recovery and alterations in motor activity patterns [9,10,11]. Those muscle dysfunctions are addressed by different physical therapy interventions, such as passive therapy, muscle strengthening, and stretching, which have shown positive results [12]. Specifically, multimodal treatments, including dry needling (DN), seem to be safe, cost-effective, and reliable to treat MTrPs [2,13,14]. According to Dunning et al. [15], DN “*encompasses the insertion of needles without injectate into, alongside, or around nerves, muscles, or connective tissues for the management of pain and dysfunction in neuromusculoskeletal conditions*”. When DN is applied directly into MTrPs, it consists of inserting a fine needle without medication into the skin, subcutaneous tissues, and muscle for eliciting LTR and disrupting mechanically MTrPs [16]. This technique should be differentiated from wet needling or injection, which uses an additional injection of analgesic substances, such as lidocaine, when performing the needling [17,18]. Among the different modalities of DN applied to the MTrPs, the most widely used is the “fast-in fast-out” or “pistoning” modality, which consists of repeated and fast entry of the needle into the MTrPs area in a fan or cone shape to elicit LTR [18].

Although there are previous reviews and meta-analyses discussing the effectiveness of DN on specific variables related to myofascial pain (highlighting its effect on reducing pain, increasing the range of motion -ROM- and reducing tone) [19,20,21], and for the treatment of MTrPs in different muscles of the lower extremity [22,23,24], a comprehensive analysis of the benefits of the DN application on triceps surae is still needed. The functional involvement of the triceps surae muscles is essential for functional activities, such as gait and balance [25]. Adequate flexibility of these muscles is related to increased dorsiflexion ROM, force production to generate elastic energy and decreased pronation upon weight acceptance [26]. For example, during landing jumps, increased dorsiflexion ROM has an impact on pre-stretching of the ankle plantar flexors, improving the utilization of elastic energy and enhancing jump function, acting as injury prevention [27].

In view of this background, our objective was to analyze the effectiveness of DN of MTrPs in the triceps surae muscles, according to the results on pain, ROM, muscle strength, muscle stiffness, and functional outcomes.

## 2. Materials and Methods

This study was performed following the guidelines of the PRISMA (Preferred Reporting Items for Systematic Review and Meta-Analysis) 2020 checklist [28] (Appendix A). In addition, this systematic review was registered in the Prospective Register of Systematic Reviews (PROSPERO), register number: CRD42021265987.

### 2.1. Search Strategy

A systematic search was performed up to February 2022 in the following scientific databases: Medline/PubMed, Physiotherapy Evidence Database (PEDro), Scopus, Cochrane Controlled Register of Trials (CENTRAL), and Web of Science. The search strategy included the following combination of terms (Appendix A): “dry needling” AND (gastrocnemius OR soleus OR calf OR “triceps surae” OR “triceps sural” OR “sural triceps”). No filters were applied by language, type of study or publication date. The literature search was performed by two authors (D.L.A. and J.A.M.M.), retrieving the potentially relevant studies. A third reviewer (C.L.M.) participated to establish a consensus when necessary. After performing the search, duplicated articles were excluded. The title and abstract were assessed and those that were not performed in humans and those that did not have the established study design were discarded. Finally, the full text of the screened articles was assessed and articles that did not meet the established selection criteria were excluded. The remaining studies were included in the systematic review.

### 2.2. Selection Criteria

As inclusion criteria, studies were selected according to the PICOS model (Population; Intervention; Comparison; Outcome/Results; Study Design) [29]: P: subjects, athletes and non-athletes, older than 18 years presenting with MTrPs in the triceps surae muscles, without any concomitant acute or chronic musculoskeletal conditions, nor other causes of neuropathic pain, such as low back radiculopathy or muscle soreness spasms; I: DN technique applied to MTrPs of the triceps surae muscles using any parameter of time of insertion, number of insertions and number of LTR in the study protocol; C: Conventional therapies, such as stretching and ischemic compression, sham DN, and no intervention; O: Results on pain, ROM, muscle strength, muscle stiffness, and functional outcomes; S: Clinical trials including an experimental group and a comparison group. There were excluded studies not performed on humans and studies using electroneedling, electroacupuncture or acupuncture or injecting any drug. In addition, the studies in which DN was applied to different muscles and the results were not reported separately for the triceps surae muscles were excluded.

### 2.3. Methodological Quality and Risk of Bias Assessment

The methodological quality of the clinical trials was assessed using the PEDro scale [30]. It consists of 11 evaluation criteria and assesses external and internal validity, as well as statistical information to interpret the results. A study scored from 9 to 11 is considered as excellent methodological quality, 6 to 8 is considered good, 4 to 5 is considered fair, and above 4 is considered poor quality [31]. Moreover, the risk of bias of each study was assessed using the Cochrane Risk of Bias tool 2.0 (RoB 2.0) [32,33]. This tool includes different questions about the risk of bias of the included studies, helping to classify among low risk, uncertain risk, or high risk.

The assessment was preformed independently by two authors (C.G.M. and C.L.M.). A third reviewer (D.L.A.) participated to establish a consensus when necessary.

### 2.4. Data Extraction

The data extracted from each article were: author/s, year of publication, intervention (treatment carried out in each group; duration, and frequency of the intervention), sample size, outcomes and measurement instruments used, and the main results (significant intergroup differences). Two independent reviewers (D.L.-A. and J.V.-M.) took part in the data extraction process and an additional reviewer (J.A.M.-M.) intervened for consensus. Furthermore, a synthesis was provided with data on the significance of the different variables measured in the included papers.

## 3. Results

### 3.1. Study Selection

The first search identified 353 potential articles; 12 met the eligibility criteria and were included in the systematic review. The screening process is reported according to the PRISMA flow diagram (Figure 1), and a detailed description of the included studies is shown in Table 1.

### 3.2. Methodological Quality and Risk of Bias Assessment

The methodological quality of the studies can be considered good according to the PEDro scale, ranging from 5 to 9 with a mean score of 6.41. Albin et al. [34] obtained the highest score (9 points) and Bandy et al. [35], Baraja-Vegas et al. [36], Pérez-Bellmunt et al. [45], Janowski et al. [43], and Cushman et al. [40] the lowest (5 points). Detailed information is shown in Table 2.

Concerning the risk of bias according to the ROB 2.0, Albin et al. [34] obtained the lowest risk, while Cushman et al. [40] obtained the highest (Figure 2). The domains with the lowest risk were missing outcome data, measurement of the outcome, and selection of the reported result (Figure 3). The domain with the highest risk was the deviations from intended interventions.

### 3.3. Population Characteristics

A total of 426 subjects were involved in the studies included in the systematic review, assuming that the studies carried out by De-Pedro et al. [37,38,39] involved the same participants. Albin et al. [34] is the study with the highest sample size, with 102 participants. Six studies included healthy non-athletic subjects [34,35,36,42,44,45], and the other six included athletes, all of them presenting MTrPs in the triceps surae muscles. From the studies in athletes, they practiced the following sports: triathlon [37,38,39], running [40], professional ballet dance [43], and any sport in which jumping, sprinting, twisting, turning, acceleration, and deceleration were important components [41]. The mean age ranged from 22.4 [45] to 42.1 [40] years, with 20–30 years being the most common age in the studies.

### 3.4. Intervention Characteristics

First, it is worth noting that all DN and comparison interventions were performed by physical therapists. All studies used the DN applied to the MTrPs, which were previously located by palpation. All the studies used the fast-in fast-out or pistoning modality, except the study by Bandy et al. [35], who only simultaneously tapped and inserted the needles into different MTrPs.

Regarding the protocol, all studies performed the DN to elicit LTR, but many studies stopped when reaching a maximum number of 10 insertions [36,37,38,39,40], and/or until the LTR were exhausted [34,36,37,38,39,40]. The remaining studies [35,41,42,43,44,45] did not describe it. All the results are based on a single session to examine the effects of DN, except for the study by Albin et al. [34], who performed two sessions.

Concerning the comparison groups, the control groups were based on sham needling in five studies [34,35,40,42,43], which consisted of performing a simulation of the DN procedure without the insertion of the needle [46]. Three studies [36,41,45] did not perform any intervention in the control group. Ischemic compression was used in three studies [37,38,39], and stretching was used in one study [44].

### 3.5. Outcome Measures

Following the selection criteria, the present systematic review focused on the following outcomes: pain, according to musculoskeletal pain [36,40,42,43] and pressure pain thresholds [37,45]; ROM [38,43,44,45]; muscle strength [34,43,45]; muscle stiffness [34,36,45]; and functional performance, such as jump height [35,41], deep squat [44], maximum power, optimal force and velocity during squat jump [41].

In addition, the included studies also reported the following outcomes: lower limb surface temperature [37,43]; adverse neural tension [42]; post-race soreness and cramps [40]; intramuscular edema [36], dynamic and static plantar pressures while standing [38], and dynamic balance [44].

The effects obtained for the different outcomes are shown in Table 3. These results suggest that DN for MTrPs in the triceps surae could have a positive impact on muscle stiffness and functional outcomes. There are inconclusive findings on musculoskeletal pain, ROM, and muscle strength. Significant results were obtained in favor of the control groups on pressure pain thresholds.

## 4. Discussion

The present systematic review aimed to analyze the effectiveness of DN of MTrPS in the triceps surae muscles in adults without any concomitant acute or chronic musculoskeletal conditions, compared with no intervention, sham needling, and conventional therapies. A total of 12 studies, involving 426 subjects, were included. To the best of our knowledge, this is the first systematic review focused on this topic. In view of the results of the included studies, DN of MTrPS in the triceps surae muscles may contribute positively to the improvement in muscle stiffness and functional outcomes. Controversial findings were found in musculoskeletal pain, ROM, and muscle strength. The results were obtained in favor of control groups for the improvement in pressure pain thresholds.

According to the obtained results, the fast-in fast-out modality of the DN technique is the most used. All the included studies used it, except of Bandy et al. [47]. Nevertheless, the protocols used in terms of time of insertion, number of insertions and number of LTR were different. In that way, there is no consensus about the best combination of parameters to obtain the MTrPS release. Therefore, we cannot recommend the protocolized application of this technique. Nevertheless, this systematic review provides an overview of the use of DN of MTrPS in the triceps surae muscles, in which the use of the fast-in fast-out modality obtained positive results in muscle stiffness and functional performance outcomes.

Regarding musculoskeletal pain, our results were based on pain perceived immediately after the intervention, obtaining controversial results. This finding coincided with those by Gattie et al. [20] and Rodríguez-Mansilla et al. [47], who found no immediate benefits for reducing pain compared with the control, sham DN or other treatments in subjects with different musculoskeletal conditions. Nevertheless, the first one found significant overall results from immediate to 12-week follow-up. This overall result is in line with two recent meta-analyses [48,49] reporting short-term benefits, but one of them used a short-term period, including from immediate follow-up to 12-weeks, and the other was from immediate follow-up to 72 h post-intervention. Therefore, our results were based on the immediate post-intervention measurements, and further research is needed to know the mid-term and long-term effects of the DN in MTrPs in the triceps surae.

Significant results were obtained in favor of the control groups for improving pressure pain thresholds. This finding agreed with the meta-analysis conducted by Gattie et al. [20], which showed no significant results immediately after DN intervention compared to no intervention, sham, and other interventions, such as ischemic compression, stretching, among others. Similar results were found by Navarro-Santana et al., who performed a meta-analysis [50] analyzing the effects of DN of MTrPs compared with no intervention and sham DN in subjects with neck pain.

In view of our findings and the previous literature, the controversial results obtained on musculoskeletal pain and no significance in pressure pain thresholds could be related to the presence of post-needling induced pain [51], the most common minor adverse event after intervention [52], which was also observed in two trials [36,40] included in this review. In this way, post-needling pain may be associated with the neuromuscular damage produced by the numerous insertions into the MTrPs [53], and, consequently, with sensitization of muscle nociceptors elicited by the inflammatory mediators [54]. This soreness could underlie the pain found after palpation in post-intervention measurement, influencing the pain perceived immediately after DN intervention [55].

Concerning the ROM outcome, we found controversial results, since there is not a clear improvement in the included studies [38,43,44,45]. On the one hand, in the literature, ROM of the lower limbs was only studied by Morihisa et al. [56], stating that DN did not have positive short-term or long-term effects on this outcome, so these findings are in line with ours. Nevertheless, this statement is based only on Huguenin et al. [57] and Mayoral et al. [58], measuring the hip internal rotation ROM and straight leg raise. Furthermore, in Mayoral et al. [58], ROM restrictions were due to the arthroplasty limitations and scar tissue formation in the knee joint capsule. On the other hand, there are recent systematic reviews with positive findings for MTrPs in the short-term [21] and in several clinical conditions, such as subacromial syndrome [59], tendinopathy [60], or stroke [61], and without effects in other conditions, such as neck pain [50]. In view of our results and the available literature, we cannot state that DN is superior to the control in improving ROM after DN in MTrPs in triceps surae. Moreover, Navarro-Santana et al. [50] supported the theories about the neurophysiological mechanism for DN approaches [14], explaining the relationship between hypoalgesic effects and improvements in ROM. In that way, Janowski et al. [43] is the only included study that evaluated both ROM and pain outcomes, and there are no significant differences in either, probably due to this relationship.

The findings obtained on muscle strength did match with the meta-analysis carried out by Mansfield et al. [62], which suggested no effects of DN therapies on muscle force production, but it was not specifically focused on triceps surae muscles in healthy subjects. In that way, although the physiological effects of the DN in muscular strength are unknown [63], improvements in physical function after DN intervention could be related to the reduction in pain [64,65]. Accordingly, only Janowski et al. [43] included both pain and muscle strength outcomes, reporting no differences between experimental and control groups. Thus, further research is needed to know if there are changes in muscle strength when DN is applied to MTrPS in triceps surae muscles.

Three studies [34,36,45] assessed the effects of DN interventions on muscle stiffness, which is commonly involved in the MTrPs pathophysiology, obtaining positive results. Our findings are related to the conclusions recently reported by Sánchez-Infante et al. [66] and Kelly et al. [67]. The first stated that DN intervention in non-active MTrPS showed lower stiffness values after 24 h and 72 h follow-up compared to the sham-DN group and baseline. The second stated that DN may cause a change post-intervention but is not maintained 24 h later. These approaches are in line with the equilibrium theory of the DN proposed by Mullins et al. [68], who stated that DN interventions could decrease spontaneous electrical activity by suppressing the mechanical effects caused by the altered length-tension relationship and the increased myofilament overlap occurring in MTrPs, resulting in improved tissue stiffness. Therefore, although those findings support our own, further research is needed to obtain solid conclusions about the long-term effects of the DN intervention, since muscle stiffness may probably be a risk factor for muscle injury [69,70].

As stated in the Introduction section, the involvement of the triceps surae muscles is relevant to functional activities [25]. In that way, our systematic review includes studies measuring functional performance through the height, maximum power, optimal force, and velocity during jump and deep squat. These two movements need an adequate flexibility of the triceps surae tissues for allowing adequate execution of the exercise [27], leading to higher test performance. Positive findings were reported for jump height and deep squat, so it seems to have some benefits on functional performance, although more randomized controlled trials are needed to confirm these results. In this way, to our knowledge, there are no published systematic reviews discussing the effect of DN on functional performance, but there are some about functional outcomes measured with functionality scales. Morihisa et al. [56] concluded that there are no improvements in functionality for DN in MTrPs in lower limbs. Gattie et al. [20] found small but significant effects for improving functional outcomes during the immediate to 12-weeks follow-up but considered several musculoskeletal conditions and measurement instruments. Therefore, there is a clear need for future research lines analyzing this issue.

In general terms, the present systematic review showed immediate positive effects on functional performance and muscle stiffness after a single session (except for the study by Albin et al. [34], which performed two sessions). According to Bandy et al. [35] and Lucas et al. [71], DN could release latent MTrPs, which affect movements via muscle activation patterns and provoke muscle weakness and fatigue, allowing greater functional performance. Therefore, we can suggest that the improvements in muscle stiffness obtained in the present review could be related to the positive results on functional performance, but this relationship should be further studied.

According to the type of participants, athletes and healthy non-athletic subjects were involved in the analyzed studies. It should be remarked that studies involving athletes [37,38,39,40,41,43] showed no positive results on any of the outcomes analyzed, except for the study conducted by Devereux et al. [41], which obtained significant results on jump height. Therefore, due to the heterogeneity of the sport modalities included, we cannot provide a categorized overview according to the influence of the characteristics of each sport modality on the outcomes. Conversely, the studies involving healthy non-athletic subjects [34,35,36,42,44,45] showed benefits on muscle stiffness [34,36,45], and functional performance, specifically, on jump height [35] and deep squat [44]. Nevertheless, muscle stiffness was not analyzed in studies involving athletes, so we cannot establish comparisons between these participants. In view of these findings, although it is well-known in the scientific literature [72] that regular physical activity is related to alterations in pain perceptions, in the studies included in this systematic review, both athletes and non-athletes obtained inconclusive results in this outcome.

### Study Limitations

The results obtained in our study could be useful in clinical practice by using DN of MTrPs in the triceps surae muscles. However, the results should be taken with caution due to the limited number of studies analyzed within each outcome. It would be advisable to unify protocols regarding the outcome assessment to obtain solid conclusions about the effectiveness of DN for improving each specific outcome, as well as to perform more randomized controlled clinical trials with higher methodological quality using larger sample sizes. Moreover, another limitation was related to the different protocols used both within the DN technique and the control groups. Another consideration is related to the single session carried out by the studies, and the measurement of only immediate and short-term effects. Therefore, the analysis of intervention programs of longer duration incorporating long-term follow-up is still needed to determine the effectiveness of DN in clinical practice. It should be highlighted that three studies analyzed in the review were performed by the same first author. These studies were probably based on data from the same participants and addressed the same topic, which could lead to a limitation, since all data analyzed by the systematic review may not be mutually independent. However, because these studies analyzed different outcomes and the review analysis has been performed in isolation for each outcome, the overall conclusion about the effect of the intervention on each outcome was not biased. Finally, due to the limited number of studies analyzing the same outcome and lack of data, a meta-analysis could not be performed, so this systematic review provides an overview of the use of DN of MTrPS in the triceps surae muscles via qualitative analysis.

## 5. Conclusions

In conclusion, although research focused on the use of DN of MTrPs in the triceps surae muscles has increased in recent years, direct scientific evidence on its effectiveness is still lacking. Nevertheless, this systematic review provides the first findings about the use of DN of MTrPs in the triceps surae muscles. Our findings suggest that this intervention could have a positive impact on muscle stiffness and functional outcomes. The positive results obtained on muscle stiffness could encourage the inclusion of DN in clinical practice, since muscle stiffness may probably be a risk factor for muscle injury. Musculoskeletal pain, ROM, and muscle strength, compared to no intervention, sham needling, and conventional therapies, obtained inconclusive results. In case of pressure pain thresholds, significant results were obtained in favor of the control groups. This may be explained by the presence of post-needling-induced pain, the most common minor adverse event after intervention, which could negatively affect the results obtained after intervention. Despite these results, they should be taken with caution because of the heterogeneity in terms of participants, DN protocols, and control groups. Therefore, well-designed research protocols are needed in future studies.

Finally, we encourage authors to carry out randomized controlled trials including follow-up measurements to determine the mid and long-term effects of DN of MTrPs in the triceps surae muscles. Further research will be necessary to integrate this intervention into clinical practice. This manuscript could be used as the base of future clinical studies and highlights the necessity of further research on the underlying mechanisms of DN.

## Figures and Tables

**Figure 1 healthcare-10-01862-f001:**
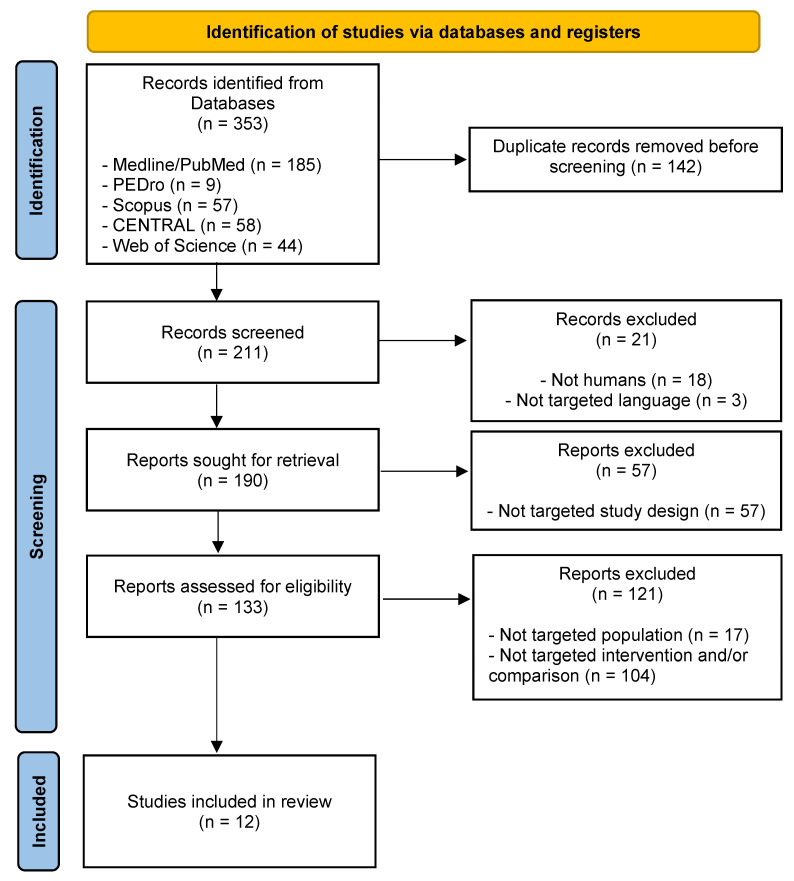
Flow diagram of the different phases of the systematic review.

**Figure 2 healthcare-10-01862-f002:**
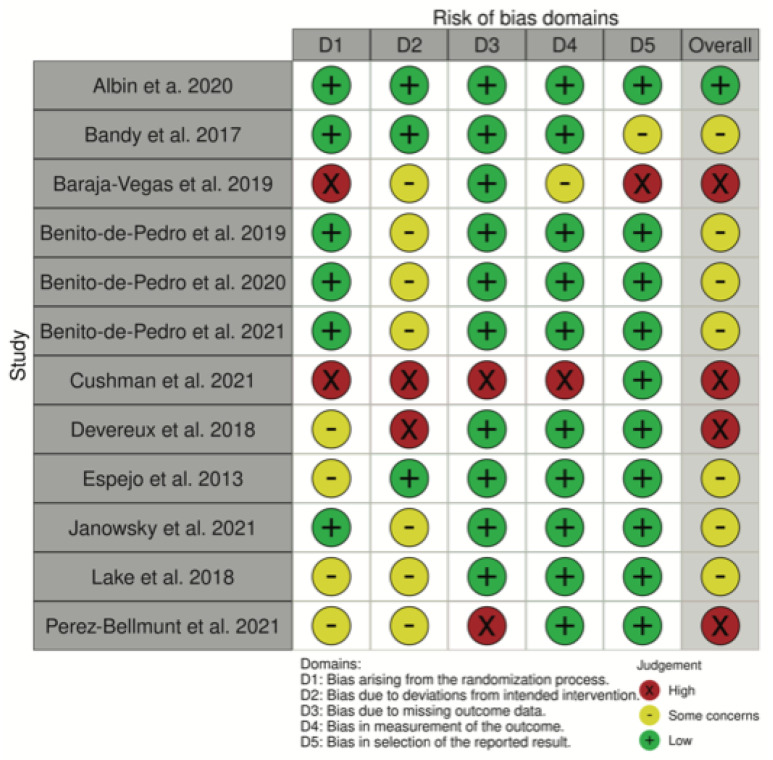
Risk of bias of the studies included in the systematic review.

**Figure 3 healthcare-10-01862-f003:**
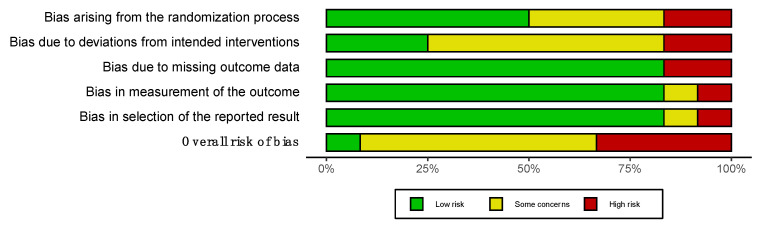
Overall risk of bias, with each category presented as percentage.

**Table 1 healthcare-10-01862-t001:** Main characteristics of the studies included in the systematic review.

Author/Year	Country	Population/Age	Type of Intervention	Intervention Dose/Method of Intervention	Outcome Measures	Measuring Instruments	Results
Albin et al. [34] 2020	United States	Non-Athletes G1: *n* = 52/25.1 ± 3.6 G2: *n* = 50/27.0 ± 5.0 N = 102/18–50 years	G1: DN G2: Sham needling	2 sessions of DN pistoning technique for 5–10 s in 3 latent MTrPs of gastrocnemius to elicit as many LTR as possible	-Muscle stiffness (resting and contracted gastrocnemius) -Muscle strength (triceps surae)	-Myoton PRO -Hand-held dynamometer	Significant improvements were found in the resting muscle stiffness at the site of the MTrPs for the DN group (*p* = 0.03).
Bandy et al. [35] 2017	United States	Non-Athletes G1: *n* = 18/ND G2: *n* = 17/ND N = 35/22.7 ± 2.4	G1: DN G2: Sham needling	1 session of DN of latent MTrPS in four sites on bilateral gastrocnemius (two at the medial head and two at the lateral head). The needles were tapped and inserted, one right after the other	-Vertical Jump height	-Chalk marks on the wall	The DN group significantly increased vertical jump eight 1.2 inches over the sham group (*p* = 0.038).
Baraja-Vegas et al. [36] 2019	Spain	Non-Athletes G1: *n* = 18 (target leg) G2: *n* = 18 (contralateral leg) N = 18/25.5 ± 5.0	G1: DN G2: Not intervention	1 session of DN in gastrocnemius latent MTrPs using the fast-in and fast-out technique during 8–10 insertions to elicit LTR	-Intramuscular edema -Muscle contractile properties -Pain	-Magnetic Resonance Imaging -Tensiomyography -11-point Numerical Pain Rating Scale	Significant changes between groups were found in the intramuscular edema for the DN group (*p* < 0.001). Significant changes between groups were found in the resting muscle stiffness with an improvement in contraction time for the DN group (*p* < 0.001).
Benito-de-Pedro et al. [37] 2019	Spain	Athletes (Triathlon) G1: *n* = 17/35.3 ± 5.4 G2: *n* = 17/33.7 ± 5.7 N = 34 (18–75 years)	G1: DN G2: Ischemic compression	1 session of deep DN in triceps surae, on latent MTrPs using the fast-in and fast-out technique to elicit LTR until the limit of tolerance of the patient or reaching a maximum number of 8 to 10 insertions	-Pressure pain thresholds -Thermographic measurement	-Wagner analog algometer -Thermographic camera with MSX technology	Statistically significant differences between groups were found in the Pressure pain threshold reduction in favor of the DN group (*p* < 0.05).
Benito-de-Pedro et al. [38] 2020	Spain	Athletes (Triathlon) G1: *n* = 17/ND G2: *n* = 17/ND N = 34 (18–75 years)	G1: DN G2: Ischemic compression	1 session of deep DN in gastrocnemius, on latent MTrPs using the fast-in and fast-out technique to elicit LTR until the LTR were exhausted, up to the limit of tolerance of the patient or reaching a maximum number of 8 to 10 insertions	-Ankle dorsiflexion ROM -Dynamic plantar pressures -Static plantar pressures	-Goniometer -Plantar pressure sensor platform with T-plate software	No significant changes between groups were found in any outcome.
Benito-de-Pedro et al. [39] 2021	Spain	Athletes (Triathlon) G1: *n* = 17/35.3 ± 5.4 G2: *n* = 17/33.7 ± 5.7 N = 34 (18–75 years)	G1: DN G2: Ischemic compression	1 session of deep DN in gastrocnemius, on latent MTrPs using the fast-in and fast-out technique to elicit LTR until the LTR were exhausted, up to the limit of tolerance of the patient or reaching a maximum number of 8 to 10 insertions	-Superficial electromyographic activity	-Electromyography	Statistically significant differences between groups were found for a reduction of superficial EMG measurements differences (%) in triathletes who train at a speed lower than 1 m/s, in favor of the DN group (*p* = 0.037).
Cushman et al. [40] 2021	United States	Athletes (Runners) G1: *n* = 28/42.1 ± 11.8 G2: *n* = 33/41.2 ± 13.1 N = 61 (>18 years)	G1: DN G2: Sham needling	1 session of DN in soleus to elicit LTR until the LTR was extinguished or reaching a maximum number of 10 insertions	-Pain (soreness) -Postrace cramps -Subjective improvement of soreness	-Numeric Pain Rating Scale -Survey	Objective pain scores showed an increase in pain of the soleus muscles at days 1 and 2 in the DN group (*p* ≤ 0.003 and *p* ≤ 0.041, respectively).
Devereux et al. [41] 2018	Ireland	Athletes (Any sport competitively in which jumping, sprinting, twisting, turning, acceleration, and deceleration were important components) G1: *n* = 10/ND G2: *n* = 10/ND G3: *n* = 10/ND G4: *n* = 10/ND N = 40/25.6 ± 5.5	G1: DN rectus femoris G2: DN medial gastrocnemius G3: DN rectus femoris + gastrocnemius G4: Not intervention	1 session of Deep DN of latent MTrPS to elicit LTR	-Jump height -Power output -Optimal force -Optimal velocity	My Jump App (iOS)	Significant improvements were found in jump height for DN medial gastrocnemius group from immediately to 48 h post-DN (*p* = 0.01).
Espejo Antúnez et al. [42] 2014	Spain	Non-Athletes G1: *n* = 23/22.4 ± 1.5 G2: *n* = 22/21.1 ± 1.3 N = 45 (>18 years)	G1: DN G2: Sham needling	1 session of DN in gastrocnemius latent MTrPs using the fast-in and fast-out technique to elicit LTR	-Adverse neural tension -Pain	-Slump neurodynamic test -Visual Analogue Scale	Significant differences were found between groups for the perceived pain in favor of the DN group (*p* < 0.01).
Janowski et al. [43] 2021	United States	Athletes (Professional ballet dancers) G1: ND G2: ND N = 11 ND	G1: DN + stretching G2: sham needling + stretching	1 session of DN in triceps surae MTrPs repeatedly moved up and down in order to elicit LTR	-Pain -Temperature -Ankle dorsiflexion ROM -Maximum muscular torque of plantar flexion	-Visual Analogue Scale -Surface thermometer -Goniometer -Biodex	No statistically significant differences between groups were found in any outcome.
Lake et al. [44] 2018	United States	Non-Athletes G1: *n* = 10/25.1 ± 2.4 G2: *n* = 10/27.1 ± 4.9 G3: *n* = 10/23.3 ± 4.8 N = 30/26.4 ± 3.1	G1: DN G2: Stretching G3: DN + stretching	G1: 1 session of DN pistoning technique (eliciting LTR) in gastrocnemius and soleus G2: 1 session of 30 s 3 times each leg G3: G1 + G2 interventions	-ROM (passive dorsiflexion, closed chain half kneeling and standing dorsiflexion) -Deep squat -Functional dorsiflexion and dynamic balance	-Inclinometer -Deep squat score -Y-Balance Test of the Lower Quarter (YBT-LQ)	Significant differences were found between groups for deep squat performance in favor of the DN group (*p* < 0.01).
Pérez-Bellmunt et al. [45] 2021	Spain	Non-Athletes G1: *n* = 25/ND G2: *n* = 25/ND N = 50/22.4 ± 8.4	G1: DN G2: Not intervention	1 session of DN in gastrocnemius latent MTrPs using the fast-in and fast-out technique to elicit LTR	-Viscoelastic properties and contractile properties -Pressure pain sensitivity -Ankle dorsiflexion ROM -Muscle strength	-MyotonPro instrument -Manual algometer/11-point Numerical Pain Rating Scale -Goniometer during lunge test -Dynamometer	Significant differences between groups were found in the lateral gastrocnemius viscoelastic properties: stiffness (*p* = 0.02), relaxation (*p* = 0.045), and creep (*p* = 0.03), in favor of the DN group. The control group showed a higher increase in pressure pain thresholds than the experimental group (*p* = 0.03).

DN: Dry Needling; G1/2: Group ½; LTR: Local Twitch Response; MTrPs: Myofascial Trigger Points.

**Table 2 healthcare-10-01862-t002:** Physiotherapy Evidence Database scores for clinical trials included in the review.

Study	1	2	3	4	5	6	7	8	9	10	11	Total
Albin et al. [34] 2020	Yes	Yes	Yes	Yes	Yes	No	Yes	Yes	Yes	Yes	Yes	9
Bandy et al. [35] 2017	Yes	Yes	No	Yes	Yes	No	No	No	No	Yes	Yes	5
Baraja-Vegas et al. [36] 2019	Yes	No	No	Yes	No	No	No	Yes	Yes	Yes	Yes	5
Benito-de-Pedro et al. [37] 2019	Yes	Yes	Yes	Yes	No	No	Yes	Yes	Yes	Yes	Yes	8
Benito-de-Pedro et al. [38] 2020	Yes	Yes	Yes	Yes	No	No	Yes	Yes	Yes	Yes	Yes	8
Benito-de-Pedro et al. [39] 2021	Yes	Yes	Yes	Yes	No	No	Yes	Yes	Yes	Yes	Yes	8
Cushman et al. [40] 2021	Yes	Yes	No	Yes	Yes	No	Yes	No	No	Yes	No	5
Devereux et al. [41] 2018	Yes	Yes	No	Yes	No	No	Yes	Yes	No	Yes	Yes	6
Espejo Antúnez et al. [42] 2014	No	Yes	No	Yes	Yes	No	Yes	Yes	No	Yes	Yes	7
Janowski et al. [43] 2021	Yes	Yes	No	No	Yes	No	Yes	No	No	Yes	Yes	5
Lake et al. [44] 2018	Yes	Yes	No	Yes	No	No	Yes	Yes	No	Yes	Yes	6
Pérez-Bellmunt et al. [45] 2021	Yes	Yes	No	Yes	No	No	Yes	No	No	Yes	Yes	5

Criterion 1 is not included in the total score, which is out of 10. Median, 6.42; range, 5 to 9. 1, eligibility criteria specified; 2, random allocation; 3, concealed allocation; 4, baseline comparability; 5, blinding of subjects; 6, blinding of therapists; 7, blinding of assessors; 8, more than 85% follow-up; 9, intention-to-treat analysis; 10, reporting of between-group statistical comparisons; 11, reporting of point measures and measures of variability.

**Table 3 healthcare-10-01862-t003:** Summary of the main effects of DN interventions compared to comparison group and associated methodological quality and risk of bias.

Author/Year	Population	Study Design	Pain	Pressure Pain Thresholds	ROM	Muscle Strength	Muscle Stiffness	Functional Performance	Overall PEDro	Overall RoB 2.0
Albin et al. [34] 2020	Non-athletes	RCT	N/A	N/A	N/A	=	+ (Resting muscle stiffness)	N/A	Excellent	Low risk
							= (Contracted muscle stiffness)			
Bandy et al. [35] 2017	Non-athletes	RCT	N/A	N/A	N/A	N/A	N/A	+ (Jump height)	Fair	Some concerns
Baraja-Vegas et al. [36] 2019	Non-athletes	CT	N/A	N/A	N/A	N/A	+	N/A	Fair	High risk
Benito-de-Pedro et al. [37] 2019	Athletes	RCT	N/A	-	N/A	N/A	N/A	N/A	Good	Some concerns
Benito-de-Pedro et al. [38] 2020	Athletes	RCT	N/A	N/A	=	N/A	N/A	N/A	Good	Some concerns
Benito-de-Pedro et al. [39] 2021	Athletes	RCT	N/A	N/A	N/A	N/A	N/A	N/A	Good	Some concerns
Cushman et al. [40] 2021	Athletes	RCT	N/A	N/A	N/A	N/A	N/A	N/A	Fair	High risk
Devereux et al. [41] 2018	Athletes	RCT	N/A	N/A	N/A	N/A	N/A	+ (Jump height)	Good	High risk
								= (Jump power output, optimal force, and velocity)		
Espejo Antúnez et al. [42] 2014	Non-athletes	RCT	+	N/A	N/A	N/A	N/A	N/A	Good	Some concerns
Janowski et al. [43] 2021	Athletes	Pilot RCT	=	N/A	=	=	N/A	N/A	Fair	Some concerns
Lake et al. [44] 2018	Non-athletes	RCT	N/A	N/A	=	N/A	N/A	+ (Deep squat)	Good	Some concerns
Pérez-Bellmunt et al. [45] 2021	Non-athletes	Within-participant RCT	N/A	-	=	=	+	N/A	Fair	High risk

RCT: Randomized controlled trial; CT: Controlled Trial; N/A: No available. Green, in favor of DN; yellow, not statistically significant; red, in favor of control/sham or other intervention.

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
