# Peer review of "Effectiveness of Dry Needling of Myofascial Trigger Points in the Triceps Surae Muscles: Systematic Review"

_healthcare, 2022, doi:10.3390/healthcare10101862_

Round 1
Reviewer 1 Report
methods
regarding intervention techniques,
Although 3 of them have the same authors, they have different techniques by different physicians.
DN techniques might not be standardized.
In general, DN repetition until finding the trigger point is largely rely on the physician. Some people
do cautiously and deeply and other not in addtion of numbers of injection.
Can the authors rule out the other possible cause of leg pain such as lumbar radiculopathy or muscle soreness spasm rather than MPS?
different groups,
In addition, athelets have different threshold of pain and different exercise level right after DN thus is should be classified differently among general population. half of the studies performed in athletes groups. what kind of sports they do? even in sports different groups have different characteristics.
In discussoin section,
the clinical effectiveness of DN shows mixed results based on each different study.
This is also due to different techniques performed by expertise. In this sense, it also shows the difficultiies of suggesting the efficacy of DN.
Can the authors suggest the possible and ideal DN methods to the readers?
in line 274~276
what is the main cause of increasing performance after DN?
is it only due to pain improvement or noteable ROM or organic change?
The authos suggest it is due to increase of muslce stiffness, but it can be also due to pain.
What is the cause and results relationship between outcome parameters such as pain, stiffness, rom etc ? One can possibly think that DN reduces pain which is a crucial factor to increas m. stiffness, rom and strength after that.
Author Response
RV: Reviewer
AA: Authors
|
RV: Methods regarding intervention techniques, Although 3 of them have the same authors, they have different techniques by different physicians. DN techniques might not be standardized. In general, DN repetition until finding the trigger point is largely rely on the physician. Some people do cautiously and deeply and other not in addtion of numbers of injection. |
|
AA: First, authors want to thank the modifications suggested by the reviewer and his/her effort to improve our manuscript. According to this suggestion, we added further information in inclusion criteria (Page 3): “DN technique applied to MTrPs of the triceps surae muscles using any parameter of time of insertion, number of insertions and number of LTR in the study protocol”. Although the protocols of the included studies were heterogeneous, the predefined purpose of this systematic review was to provide a wide overview of using Dry needling applied to MTrPS of the triceps surae muscles. |
|
RV: Methods Can the authors rule out the other possible cause of leg pain such as lumbar radiculopathy or muscle soreness spasm rather than MPS? |
|
AA: Following the reviewer’s recommendation, the sentence was rewritten including other possible causes of leg pain Page 3: “P: subjects, athletes and non-athletes, older than 18 years presenting with MTrPs in the triceps surae muscles, without any concomitant acute or chronic musculoskeletal conditions, nor other causes of neuropathic pain, such as low back radiculopathy or muscle soreness spasms” |
|
RV: different groups, In addition, athelets have different threshold of pain and different exercise level right after DN thus is should be classified differently among general population. half of the studies performed in athletes groups. what kind of sports they do? even in sports different groups have different characteristics. |
|
AA: According to the reviewer’s suggestion, we added the information about the specific sport of each article in the Table 1. In the Result section and subsection 3.3 (Page 8), information about the sport modalities were included: “From the studies in athletes, they practiced the following sports: triathlon [38–40], running [41], professional ballet dance [44], and any sport in which jumping, sprinting, twisting, turning, acceleration, and deceleration were important components [42] . Finally, in the Discussion section (Page 13), the following sentences were added: “Therefore, due to the heterogeneity of the sport modalities included, we cannot provide a categorized overview according to the influence of the characteristics of each sport modality on the outcomes.” “In view of these findings, although is well-known in the scientific literature [74] that the regular physical activity is related to alterations in pain perceptions, in the studies included in this systematic review, both athletes and non-athletes obtained inconclusive results in this outcome.” The reference [74] were added according to this comment: Tesarz, J.; Schuster, A.K.; Hartmann, M.; Gerhardt, A.; Eich, W. Pain Perception in Athletes Compared to Normally Active Controls: A Systematic Review with Meta-Analysis. Pain 2012, 153, 1253–1262, doi:10.1016/J.PAIN.2012.03.005. |
|
RV: In discussion section, the clinical effectiveness of DN shows mixed results based on each different study. This is also due to different techniques performed by expertise. In this sense, it also shows the difficulties of suggesting the efficacy of DN. Can the authors suggest the possible and ideal DN methods to the readers? |
|
AA: In order to solve this question, a new paragraph in the Discussion section (Page 11) was added to state our position about the use of dry needling of myofascial trigger points in the triceps surae muscles: “According to the obtained results, the fast-in fast-out modality of the DN technique is the most used. All the included studies used it, except of Bandy et al. [48]. Nevertheless, the protocols used in term of time of insertion, number of insertions and number of LTR were different. In that way, there is no consensus about the best combination of parameters to obtain the MTrPS release. Therefore, we cannot recommend a protocolized application of this technique. Nevertheless, this systematic review provides an overview of the use of DN of MTrPS in the triceps surae muscles in which the use of the fast-in fast-out modality obtained positive results in muscle stiffness and functional performance outcomes.”
|
|
RV: In lines 274~276 what is the main cause of increasing performance after DN? is it only due to pain improvement or noteable ROM or organic change? The authos suggest it is due to increase of muslce stiffness, but it can be also due to pain. What is the cause and results relationship between outcome parameters such as pain, stiffness, rom etc ? One can possibly think that DN reduces pain which is a crucial factor to increas m. stiffness, rom and strength after that. |
|
AA: Following the reviewer’s suggestion, we agree with the influence of the pain in the improvement of the rest of parameters. Nevertheless, the included studies did not reported improvements in musculoskeletal pain and pain thresholds and the physiological mechanisms that support the Dry needling technique remain unknown (doi:10.1007/s11916-013-0348-5). Therefore, we consider it is not adequate to establish a relationship between pain and muscle stiffness and functional performance outcomes, which obtained positive results.
In that way, both findings were explained along the discussion. First, we provided an explanation about the no positive results obtained in pain (Page 11): “In view of our findings and the previous literature, the controversial results obtained on musculoskeletal pain and no significance in pressure pain thresholds could be related to the presence of post-needling induced pain [53], the most common minor adverse event after intervention [54], which was also observed in two trials [37,41]included in this review. In this way, post-needling pain may be associated to the neuromuscular damage produced by the numerous insertions into the MTrPs [55], and, consequently, with sensitization of muscle nociceptors elicited by the inflammatory mediators [56]. This soreness could underlie the pain found after palpation in the post-intervention measurement, influencing the pain perceived immediately after DN intervention [57].”. Second, we provided a possible explanation about the influence of the muscle stiffness improvement on functional performance (Page 13): “According to Bandy et al. [36] and Lucas et al. [73], DN could release latent MTrPs, which affect movements via muscle activation patterns and provoke muscle weakness and fatigue, allowing greater functional performance. Therefore, we can suggest that the improvements in muscle stiffness obtained in the present review could be related to the positive results on functional performance, but this relationship should be further studied.”
Cagnie, B.; Dewitte, V.; Barbe, T.; Timmermans, F.; Delrue, N.; Meeus, M. Physiologic Effects of Dry Needling. Curr Pain Headache Rep 2013, 17, 348, doi:10.1007/s11916-013-0348-5. |
Reviewer 2 Report
Introduction :
32-34 : the definition of a "trigger point" - ref 4 - ( in Spanish !!) is a clinical approach where no objective parameter exists to make a distinction between a muscle with or without a trigger point.
34 - 39 : The clinical findings associated with...(ref 5-6 : Simons et al) is based on an opinion from the authors of a book (Simons, Travell and Simons..ref 2 : " the trigger point manual ")
44-47 : definition of dry needling based on ref 15 : a description of a technique, no validation of a technique.
line 48 : ref 16 (Hong) : this is a modified technique, based again on Simons, Travell of INJECTION and no "dry needling"
159-168 : the "interventions" are not at all comparable .... and what is "sham needling" ?
line 42 : reference with one of the authors ref 11 : Gomez.. .... Luque-Moreno C : Effectiveness of stretching ibn Post stroke spasticity and range of motion .. , this has nothing to do with the subject of dry needling.
My ethical conflict with this article is also the fact that "dry needling" is a technique where the skin is penetrated, this is a "medical act", most of the authors that are referred to are physiotherapists. The medical community is still not accepting this treatment performed by non-medical practitioners. lines 154-156 : all interventions were performed by physiotherapists...used palpation to find the MTrPs ....
Conclusion : 326 -328 :
"...this systematic review provides the first evidence-based findings..." : a review is NOT yet an evidence, the first randomised trial with a definition of what you want to treat and how you treat it has still to be published in my opinion.
Author Response
RV: Reviewer
AA: Authors
|
RV: Introduction: 32-34 : the definition of a "trigger point" - ref 4 - ( in Spanish !!) is a clinical approach where no objective parameter exists to make a distinction between a muscle with or without a trigger point. |
|
AA: First, authors want to thank the reviewer’s suggestions and his/her effort to improve our manuscript. Following this first recommendation, the paragraph was rewritten, as follows: Page 1: “MTrPs are clinically defined as a hyperirritable nodule of spot tenderness located in a taut band of skeletal muscle which is tender and palpable through the physical examination [4]. A recent study carried out by Li et al. [5] stated that the diagnostic of MTrPs is mostly based on the presence of three criteria, either as a stand-alone or combined: spot tenderness, referred pain and local twitch response (LTR). The use of these criteria combined provide a more reliable diagnostic, as it is known that the reliability of each criterion is associated with the analyzed muscle [6]. In addition to physical examination, taut bands can be objectively characterized by magnetic resonance [7], and the irritability caused by MTrPs can be showed by electromyography [3,8]. Nevertheless, an objective standard diagnostic of MTrPs is still needed [5].” The previous reference 4 was removed, and six references were added to the manuscript in order to provide a more comprehensive approach about the definition and diagnosis of MTrPs: 3. Celik, D.; Mutlu, E.K. Clinical Implication of Latent Myofascial Trigger Point Topical Collection on Myofascial Pain. Curr Pain Headache Rep 2013, 17, doi:10.1007/s11916-013-0353-8. 4. Shah, J.P.; Thaker, N.; Heimur, J.; Aredo, J. v.; Sikdar, S.; Gerber, L. Myofascial Trigger Points Then and Now: A Historical and Scientific Perspective. PM and R 2015, 7, 746–761. 5. Li, L.; Stoop, R.; Clijsen, R.; Hohenauer, E.; Fernández-De-Las-Peñas, C.; Huang, Q.; Barbero, M. Criteria Used for the Diagnosis of Myofascial Trigger Points in Clinical Trials on Physical Therapy: Updated Systematic Review. Clinical Journal of Pain 2020, 36, 955–967. 6. Myburgh, C.; Larsen, A.H.; Hartvigsen, J. A Systematic, Critical Review of Manual Palpation for Identifying Myofascial Trigger Points: Evidence and Clinical Significance. Arch Phys Med Rehabil 2008, 89, 1169–1176, doi:10.1016/J.APMR.2007.12.033. 7. Chen, Q.; Basford, J.; An, K.N. Ability of Magnetic Resonance Elastography to Assess Taut Bands. Clinical Biomechanics 2008, 23, 623–629, doi:10.1016/j.clinbiomech.2007.12.002. 8. Kuan, T.S.; Hsieh, Y.L.; Chen, S.M.; Chen, J.T.; Yen, W.C.; Hong, C.Z. The Myofascial Trigger Point Region: Correlation between the Degree of Irritability and the Prevalence of Endplate Noise. Am J Phys Med Rehabil 2007, 86, 183–189, doi:10.1097/PHM.0b013e3180320ea7. |
|
RV: Introduction: 34 - 39 : The clinical findings associated with...(ref 5-6 : Simons et al) is based on an opinion from the authors of a book (Simons, Travell and Simons..ref 2 : " the trigger point manual ") |
|
AA: Following the reviewer’s suggestion, the previous references were removed and the paragraph included in the previous comments also add information in order to provide a more comprehensive approach about the diagnosis of MTrPs:
Page 1: “MTrPs are clinically defined as a hyperirritable nodule of spot tenderness located in a taut band of skeletal muscle which is tender and palpable through the physical examination [4]. A recent study carried out by Li et al. [5] stated that the diagnostic of MTrPs is mostly based on the presence of three criteria, either as a stand-alone or combined: spot tenderness, referred pain and local twitch response (LTR). The use of these criteria combined provide a more reliable diagnostic, as it is known that the reliability of each criterion is associated with the analyzed muscle [6]. In addition to physical examination, taut bands can be objectively characterized by magnetic resonance [7], and the irritability caused by MTrPs can be showed by electromyography [3,8]. Nevertheless, an objective standard diagnostic of MTrPs is still needed [5].”
|
|
RV: Introduction: 44-47 : definition of dry needling based on ref 15 : a description of a technique, no validation of a technique. |
|
AA: In order to add new sources of information about the dry needling technique, the definition was rewritten and new references were added: Page 2: “According to Dunning et al. [16], DN “encompasses the insertion of needles without injectate into, alongside, or around nerves, muscles, or connective tissues for the management of pain and dysfunction in neuromusculoskeletal conditions”. When DN is applied directly into MTrPs, it consists of inserting a fine needle without medication into the skin, subcutaneous tissues, and muscle for eliciting LTR and disrupting mechanically MTrPs [17]. This technique should be differentiated from the wet needling or injection, that uses an additional injection of analgesic substances, such as lidocaine, when performing the needling [18,19].” The previous reference 15 was removed, and three new references were added to support the definition: 16. Dunning, J.; Butts, R.; Mourad, F.; Young, I.; Flannagan, S.; Perreault, T. Dry Needling: A Literature Review with Implications for Clinical Practice Guidelines. Physical Therapy Reviews 2014, 19, 252–265, doi:10.1179/108331913x13844245102034. 17. Lew, J.; Kim, J.; Nair, P. Comparison of Dry Needling and Trigger Point Manual Therapy in Patients with Neck and Upper Back Myofascial Pain Syndrome: A Systematic Review and Meta-Analysis. Journal of Manual and Manipulative Therapy 2021, 29, 136–146, doi:10.1080/10669817.2020.1822618. 18. Zha, M.; Chaffee, K.; Alsarraj, J. Trigger Point Injections and Dry Needling Can Be Effective in Treating Long COVID Syndrome-Related Myalgia: A Case Report. J Med Case Rep 2022, 16, doi:10.1186/s13256-021-03239-w. 19. 19. Perreault, T.; Dunning, J.; Butts, R. The Local Twitch Response during Trigger Point Dry Needling: Is It Necessary for Successful Outcomes? J Bodyw Mov Ther 2017, 21, 940–947, doi:10.1016/j.jbmt.2017.03.008.
|
|
RV: Introduction: line 48 : ref 16 (Hong) : this is a modified technique, based again on Simons, Travell of INJECTION and no "dry needling" |
|
AA: According to the reviewer’s recommendation, the sentence was replaced. Page 2: “This technique should be differentiated from the wet needling or injection, that uses an additional injection of analgesic substances, such as lidocaine, when performing the needling [18,19]. Among the different modalities of DN applied to the MTrPs, the most widely used is the “fast-in fast-out” or “pistoning” modality, which consists of repeated and fast entry of the needle into the MTrPs area in a fan or cone shape to elicit LTR [19].” The previous references 16 and 17 were removed and two new references were added: 18. Zha, M.; Chaffee, K.; Alsarraj, J. Trigger Point Injections and Dry Needling Can Be Effective in Treating Long COVID Syndrome-Related Myalgia: A Case Report. J Med Case Rep 2022, 16, doi:10.1186/s13256-021-03239-w. 19. Perreault, T.; Dunning, J.; Butts, R. The Local Twitch Response during Trigger Point Dry Needling: Is It Necessary for Successful Outcomes? J Bodyw Mov Ther 2017, 21, 940–947, doi:10.1016/j.jbmt.2017.03.008. |
|
RV: 159-168 : the "interventions" are not at all comparable .... and what is "sham needling" ? |
|
AA: Following the reviewer’s suggestion, the “3.4. Intervention characteristics” section was restructured, and specific information about the sham needling was added. Page 9: “All studies used the DN applied to the MTrPs which were previously located by palpation. All the studies used the fast-in fast-out or pistoning modality, except the study by Bandy et al. [36], who only simultaneously tapped and inserted the needles into different MTrPs. Regarding the protocol, all studies performed the DN to elicit LTR, but many studies stopped when reaching a maximum number of 10 insertions [37–41], and/or until the LTR were exhausted [35,37–41]. The remaining studies [36,42–46] did not describe it . All the results are based on a single session to examine the effects of DN, except for the study by Albin et al. [35], who performed two sessions. Concerning the comparison groups, the control groups were based on sham needling in five studies [35,36,41,43,44], which consisted of performing a simulation of the DN procedure without the insertion of the needle [47]. Three studies [37,42,46]did not perform any intervention in the control group. Ischemic compression was used in three studies [38–40], and stretching was used in one study [45].” A new reference was added: 47. Braithwaite, F.A.; Walters, J.L.; Li, L.S.K.; Moseley, G.L.; Williams, M.T.; Mcevoy, M.P.; Fa, B.; Jl, W.; Lsk, L.; Gl, M.; et al. Blinding Strategies in Dry Needling Trials: Systematic Review and Meta-Analysis Background. Blinding of Participants and Therapists in Trials of Physical Interventions. Phys Ther 2019, 99, 1461–1480. In addition, a new statement was added to the “study limitations” section: Page 13: “Moreover, another limitation was related to the different protocols used both within the DN technique and the control groups.” |
|
RV: Introduction: line 42 : reference with one of the authors ref 11 : Gomez.. .... Luque-Moreno C : Effectiveness of stretching ibn Post stroke spasticity and range of motion .. , this has nothing to do with the subject of dry needling. |
|
AA: According to the reviewer’s recommendation, the reference was removed. |
|
RV: My ethical conflict with this article is also the fact that "dry needling" is a technique where the skin is penetrated, this is a "medical act", most of the authors that are referred to are physiotherapists. The medical community is still not accepting this treatment performed by non-medical practitioners. lines 154-156 : all interventions were performed by physiotherapists...used palpation to find the MTrPs .... |
|
AA: First of all, the authors appreciate that the reviewer explicitly states his/her conflict with the use of dry needling. We are aware of the conflicts that may arise between the competencies of medical and physical therapist professionals in this case. In our opinion, this lack of consensus may be due to differences in the regulation of professional practice between different countries. This confrontation regarding the inclusion of dry needling in clinical practice by different professionals has been analyzed by different authors, such as Dommerholt et al.: “Physical therapists around the world practice TrP-DN as part of their clinical practice and use the technique in combination with other physical therapy interventions. TrP-DN falls within the scope of physical therapy practice in many countries, including Canada, Chile, Ireland, the Netherlands, South Africa, Spain, and the United Kingdom. In 2002, two Dutch medical courts ruled that TrP-DN is within the scope of physical therapy practice in the Netherlands even though the Royal Dutch Physical Therapy Association has expressed the opinion that TrP-DN should not be part of physical therapy practice2-4. Of the approximately 9,000 physical therapists in South Africa, over 75% are estimated to employ the technique at least once daily (Stavrou, per sonal communication, 2006). Physical therapy continuing education programs in TrP-DN in Ireland, Switzerland, and Spain are popular among physical therapists. In Spain, several universities offer academic and specialist certification programs featuring TrP-DN as an integral component of invasive physical therapy5. In the United States (US) and Australia, TrP-DN is not commonly included in physical therapy entry-level educational curricula or post-graduate continuing education programs. Relatively few physical therapists in those two countries have received training in and employ the technique. The only known US physical therapy academic program that includes course work in TrP-DN is the entry-level doctorate of physical therapy curriculum at Georgia State University (Donnelly, personal communication, 2006). However, the physical therapy state boards of Colorado, Georgia, Kentucky, Maryland, New Hamp shire, New Mexico, South Carolina, and Virginia have determined in recent years that TrP-DN does fall within the scope of physical therapy in those states. Several other state boards are currently reviewing whether dry needling should fall within the scope of physical therapy practice, and the Director of Regulations of the State of Colorado has issued a specific “Director’s Policy on Intramuscular Stimulation” (Table 1)6...” - Dommerholt, J.; Mayoral del Moral, O.; Gröbli, C. Trigger point dry needling. Journal of Manual & Manipulative Therapy 2006, 14: 70E-87E, doi:10.1179/jmt.2006.14.4.70E.
In addition, this topic was also addressed by Dunning et al. in their section: “Narrowly Focused Position Statements by State Boards of Physical Therapy”
- Dunning, J.; Butts, R.; Mourad, F.; Young, I.; Flannagan, S.; Perreault, T. Dry Needling: A Literature Review with Implications for Clinical Practice Guidelines. Physical Therapy Reviews 2014, 19, 252–265, doi:10.1179/108331913x13844245102034. |
|
RV: Conclusion : 326 -328 : "...this systematic review provides the first evidence-based findings..." : a review is NOT yet an evidence, the first randomised trial with a definition of what you want to treat and how you treat it has still to be published in my opinion. |
|
AA: Following the reviewer’s recommendation, the "evidence-based" statement was removed. |
Round 2
Reviewer 2 Report
Table 1 :
There are 3 publications from "Benito-de-Pedro et al" where 17 patients and 17 controls are included, in the publiactions from 2019 (38) and 2021 (40) they are probably the same 17 patients, exactly the same age. In the publication from 2020 (39) there are again these 17 patients ? Are they listed as 17 or 3 x 17 in the total count of 494 patients ?
I still have doubts about the added value of this review. The relation between trigger points and clinical symptoms is still not proven and certainly not the cure of symptoms through dry needling. The "patients"group aimed in these studies is so different, same about the controls. The technique is not the same, the therapists are of different qualifications...The only conclusion can be that a well designed research protocol is needed.
Author Response
Response to Reviewers. Round 2 (healthcare-1897402).
Title: Effectiveness of Dry Needling of Myofascial Trigger Points in the Triceps Surae Muscles: Systematic Review Dear Reviewer,
the authors would like to thank you for your comments. A detailed response to the comments is provided below.
Reviewer 1
RV: Reviewer
AA: Authors
|
RV: able 1 : There are 3 publications from "Benito-de-Pedro et al" where 17 patients and 17 controls are included, in the publiactions from 2019 (38) and 2021 (40) they are probably the same 17 patients, exactly the same age. In the publication from 2020 (39) there are again these 17 patients ? Are they listed as 17 or 3 x 17 in the total count of 494 patients ? |
|
AA: First, the authors would like to thank the reviewer for his/her consideration. In order to better specify the number of subjects who participated in the studies included in the systematic review, the following informations was added in the section “3.3. Population characteristics”. “A total of 426 subjects were involved in the studies included in the systematic review, assuming that the studies carried out by De-Pedro et al. [38–40] involved the same participants. Albin et al. [35] is the study with the highest sample size with 102 participants.”
|
|
RV: I still have doubts about the added value of this review. The relation between trigger points and clinical symptoms is still not proven and certainly not the cure of symptoms through dry needling. The "patients"group aimed in these studies is so different, same about the controls. The technique is not the same, the therapists are of different qualifications...The only conclusion can be that a well designed research protocol is needed. |
|
AA: The authors agree with the reviewer's remarks on the deficiencies that currently exist with respect to the use of dry needling. Nontheless, the scientific literature about this technique is increasing and it is hihgly used in clinical setting in those countries in where it is allowed. Although the trigger point theory and the physiological mechanisms of the application of dry needling remain controversial, the present systematic review provides an overview about the current literature on its aplication in triceps surae muscles. The authors crearly highlighted the pros and cons of using this technique, showing the methodological drawbacks and implications to the clinical practice. Therefore, this manuscript can be the basis for future research in which these concerns can be addressed. In that way, a statement was added in the section “5. Conclusions” to clearly show the limited current evidence of the dry needling applied to the myofascial trigger points in triceps surae: “Despite these results, they should be taken with caution because of the heterogeneity in terms of participants, DN protocols, and control groups. Therefore, well-designed research protocols are needed in future studies.”
|